# *LPCAT1-TERT* fusions are uniquely recurrent in epithelioid trophoblastic tumors and positively regulate cell growth

Gavin R. Oliver[1,2]*, Sofia Marcano-Bonilla[1,2], Jonathan Quist[1,2], Ezequiel J. Tolosa[3], Eriko Iguchi[3], Amy A. Swanson[4], Nicole L. Hoppman[4], Tanya Schwab[5], Ashley Sigafoos[5], Naresh Prodduturi[1,2], Jesse S. Voss[4], Shannon M. Knight[4], Jin Zhang[4], Numrah Fadra[1,2], Raul Urrutia[2], Michael Zimmerman[1,2], Jan B. Egan[1], Anthony G. Bilyeu[4], Jin Jen[4], Ema Veras[6], Rema'a Al-Safi[7], Matthew Block[1], Sarah Kerr[4], Martin E. Fernandez-Zapico[3], John K. Schoolmeester[4], Eric W. Klee[1,2]

1 Center for Individualized Medicine, Mayo Clinic, Rochester, Minnesota, United States of America,
2 Division of Health Sciences Research, Mayo Clinic, Rochester, Minnesota, United States of America,
3 Division of Oncology Research, Schulze Center for Novel Therapeutics, Department of Oncology, Mayo Clinic, Rochester, Minnesota, United States of America, 4 Department of Laboratory Medicine and Pathology, Mayo Clinic, Rochester, Minnesota, United States of America, 5 Department of Biochemistry and Molecular Biology, Mayo Clinic, Rochester, Minnesota, United States of America, 6 Sibley Memorial Hospital, Johns Hopkins Medicine, Washington, DC, United States of America, 7 Gynecologic Pathology, Histopathology and Cytology Units, Maternity Hospital, Kuwait City, Kuwait

* oliver.gavin@mayo.edu

**Data Availability Statement:** Copy number array data have been deposited in ArrayExpress with accession E-MTAB-10303 while sequencing data

## Abstract

Gestational trophoblastic disease (GTD) is a heterogeneous group of lesions arising from placental tissue. Epithelioid trophoblastic tumor (ETT), derived from chorionic-type trophoblast, is the rarest form of GTD with only approximately 130 cases described in the literature. Due to its morphologic mimicry of epithelioid smooth muscle tumors and carcinoma, ETT can be misdiagnosed. To date, molecular characterization of ETTs is lacking. Furthermore, ETT is difficult to treat when disease spreads beyond the uterus. Here using RNA-Seq analysis in a cohort of ETTs and other gestational trophoblastic lesions we describe the discovery of *LPCAT1-TERT* fusion transcripts that occur in ETTs and coincide with underlying genomic deletions. Through cell-growth assays we demonstrate that LPCAT1-TERT fusion proteins can positively modulate cell proliferation and therefore may represent future treatment targets. Furthermore, we demonstrate that *TERT* upregulation appears to be a characteristic of ETTs, even in the absence of *LPCAT1-TERT* fusions, and that it appears linked to copy number gains of chromosome 5. No evidence of *TERT* upregulation was identified in other trophoblastic lesions tested, including placental site trophoblastic tumors and placental site nodules, which are thought to be the benign chorionic-type trophoblast counterpart to ETT. These findings indicate that *LPCAT1-TERT* fusions and copy-number driven *TERT* activation may represent novel markers for ETT, with the potential to improve the diagnosis, treatment, and outcome for women with this rare form of GTD.

have been deposited under accession E-MTAB-10321.

**Funding:** We wish to acknowledge Mayo Clinic Center for Individualized Medicine and the Department of Laboratory Medicineand Pathology for supporting this study. GTEx data used for the analyses described in this manuscript were obtained from dbGaP accession number phs000424.v7.p2. The Genotype-Tissue Expression (GTEx) Project was supported by the Common Fund of the Office of the Director of the National Institutes of Health, and by NCI, NHGRI, NHLBI, NIDA, NIMH, and NINDS.

**Competing interests:** The authors have declared that no competing interests exist.

## Introduction

Gestational trophoblastic disease (GTD) is a heterogeneous group of lesions that includes both neoplastic and non-neoplastic entities. As defined by the 2014 WHO Classification of Tumors of Female Reproductive Organs [1], choriocarcinoma, placental site trophoblastic tumor (PSTT) and epithelioid trophoblastic tumor (ETT) encompass the neoplasms [2], whereas exaggerated implantation/placental site and placental site nodule (PSN) are non-neoplastic counterparts. Complete, partial and invasive hydatidiform moles represent abnormally developed non-neoplastic trophoblastic proliferations that carry potential for neoplastic transformation [3].

ETT, the rarest form of GTD [4, 5], is composed of chorionic-type intermediate trophoblast that has potential for metastasis. Although PSN is also composed of chorionic-type intermediate trophoblast, it is benign. While choriocarcinoma is more readily distinguishable, ETT, PSTT and PSN may present overlapping clinical and pathological features with associated diagnostic challenges [6]. ETT can be misdiagnosed [5, 7] as PSN, PSTT, clear cell carcinoma, or several other tumor types. Mixed histologies are also observed [6], and recently lesions possessing features between those of PSN and ETT have been classified as atypical PSNs [3]. Atypical PSNs are considered intermediate lesions in the spectrum of PSN and ETT. Furthermore, it has been proposed that ETT and PSTT might evolve from a previous PSN [8–10].

While the clinical course of ETT is difficult to predict, the risk of metastasis at the time of diagnosis is 25% [11], and the overall mortality rate is estimated to be as high as 24% [12]. Incorrectly diagnosing ETT is undesirable since hysterectomy can be curative for uterine-confined disease. ETT often shows a poor response to chemotherapy [3], which is generally reserved for metastatic disease, or disease presenting greater than four years following an antecedent pregnancy [3, 11]. This relative chemotherapy resistance dictates that ETT accounts disproportionately for GTD-related mortality, creating a need for novel therapeutic modalities to improve outcomes of women with advanced disease [11].

Here, we describe the application of RNA-Seq-based fusion transcript detection in GTD. Through profiling nine cases of GTD comprising ETT, PSTT and PSN, we identify and experimentally confirm the presence of *LPCAT1-TERT* fusion transcripts that appear to uniquely reoccur in ETT and are caused by genomic deletions. *TERT* is a well-established oncogene, whose expression is inactivated in most normal tissues but detectable in the majority of tumors [13]. Conversely *LPCAT1* is ubiquitously expressed in normal tissues and encodes a protein with acyltransferase [14] and acetyltransferase activities [15] with proposed roles in respiratory physiology [14] and regulation of lipid droplet size and quantity [16]. While not universally recognized as an oncogene, growing evidence links *LPCAT1* overexpression to cancer progression, metastasis and recurrence in oral, kidney, breast, gastric, and lung cancers [17, 18].

We demonstrate that LPCAT1-TERT fusion proteins promote cell growth in surrogate non-transformed 293T cells and could therefore represent relatively early events in ETT pathogenesis. Further, we show that copy number gains of chromosome 5 accompany *TERT* upregulation in ETT. Together, our findings define novel and unique features that may participate in ETT pathogenesis and indicate the potential for novel diagnostic or therapeutic considerations.

## Materials and methods

### Ethics statement

The final study was reviewed and approved by the Mayo Clinic Institutional Review Board. One individual (case ETT-1) was originally enrolled as part of a prospective study and provided written consent. All other cases involved fully anonymized, archival samples, and the need for consent was waived by the Institutional Review Board at their respective institution.

## Histopathology

All tissue specimens were subjected to standard macroscopic and histological examinations. Tissue sections were processed routinely for morphologic assessment: sections were fixed in neutral-buffered formalin, processed, embedded in paraffin, sectioned and stained with hematoxylin and eosin. GTDs were diagnosed by pathologists at participating centers prior to final confirmation utilizing independent review by two gynecologic pathologists (JKS and SEK), following WHO classification guidelines [1].

## Sample extraction

Sections of formalin-fixed paraffin-embedded (FFPE) tissue were freshly cut for nucleic acid extraction. DNA and RNA were extracted from FFPE tissue sections using the AllPrep DNA/ RNA FFPE Kit (Qiagen, Netherlands) according to the manufacturer's protocol. DNA was eluted in 30–50 μl ATE Buffer and RNA was eluted in 20–30 μl RNase-free water. DNA and RNA concentrations were quantified using Qubit fluorometry (Invitrogen, Carlsbad, CA).

## RNA sequencing

Sequencing libraries were prepared according to manufacturer's instructions for either the TruSeq RNA Sample Prep Kit v2 or the TruSeq RNA Access Library Prep Kit (Illumina, San Diego, CA). Library concentration and size distribution were initially determined using an Agilent Bioanalyzer DNA 1000 chip (Santa Clara, CA), and Qubit fluorometry (Invitrogen, Carlsbad, CA) was performed to confirm concentration. Paired-end 101-basepair reads were sequenced on an Illumina HiSeq 2500 using the TruSeq Rapid SBS sequencing kit version 1 and HCS version 2.0.12.0 data collection software. A median of approximately 175 million reads was generated per sample.

## RNA expression analysis

Reads were aligned to the human genome (hg19) and transcriptome using Tophat2 [19] running Bowtie (v1) [20]. Gene level read counts were generated using HtSeq [21] and BedTools [22] respectively. Normalization was performed using the median of ratios method implemented by DESeq2 (v1.26.0) [23].

## Fusion detection and analysis

Candidate fusion events were detected using TopHat Fusion (TopHat release 2.1.0) [24] with all default filters disabled to maximize sensitivity. To control for common events and recurrent artifacts, we compared putative fusion candidates to a database generated using tumor and normal samples from our own institution, the Illumina Human BodyMap, The Cancer Genome Atlas Fusion Database [25], and the Genotype-Tissue Expression (GTEx) project (dbGaP accession phs000424.v7.p2) [26]. GTEx consisted of approximately 8200 RNA-Seq samples from 549 unique individuals and 52 tissue types following QC.

## DNA copy number analysis

Chromosomal microarray (CMA) was performed on DNA extracted from FFPE tissue using the OncoScan CNV Plus assay (Thermo Fisher Scientific, Waltham, MA) in a clinical laboratory and according to the manufacturer's protocol. CMA data were analyzed using ChAS software version 3.3 (Thermo Fisher Scientific, Waltham, MA) and interpreted by a board-certified clinical cytogeneticist.

## *TERT* promoter analysis

DNA mutation analysis was conducted within the *TERT* promoter region (hg19 chr5:1295170–1295296) for all samples. The *TERT* promoter region was amplified using gene specific primer sequences with Illumina adapter sequence on the 5' end (F- AGTTCAGACG TGTGCTCTTCCGATCTCGTCCTGCCCCTTCACCT R- TCCCTACACGACGCTCTTCCGATC TAGCGCTGCCTGAAACTCG) and the KAPA Hi-Fi Hotstart ReadyMix master mix per manufacturer protocol. After Ampure XP purification, a 2^nd round of PCR was performed to add a patient specific barcode and an Illumina specific adapter sequence. After final purification, samples were loaded onto an Illumina MiSeq instrument. Data were processed using a custom bioinformatics pipeline and any *TERT* promoter mutations with 5% or greater mutant allele frequency were reported.

## Fusion expression validation in primary ETT

*LPCAT1-TERT* fusion expression was validated by RT-qPCR as follows: RNA was extracted from FFPE ETT or adjacent normal tissue sections using the RNA FFPE Kit (Qiagen, Netherlands) according to the manufacturer's protocol. Two micrograms of RNA were used to generate cDNA using High-Capacity cDNA Reverse Transcription Kit (Applied Biosystems). Then, the cDNA was amplified by real-time PCR. Samples were prepared with PerfeCTa SYBR Green FastMix (Quanta BioSciences Inc) and the following primer sets: *LPCAT1-TERT* sense: 5'-CGCCTCACTCGTCCTACTTC-3', antisense 5'-TTGCAACTTGCTCCAGACAC-3 and *18S*, sense 5'-AACCCGTTGAACCCCATTCGTGAT-3', antisense 5'-AGTCAAGTTCGACC GTCTTCTCAG-3'. Furthermore, 5 µl of the cDNA was amplified by regular PCR using the *LPCAT1-TERT* primers already described, and TAKARA Kit according to the manufacturer's protocol. The PCR products were resolved in a 2% agarose gel and the 278 bp bands were extracted by Gel/PCR DNA Fragment Extraction Kit (MicSci) and Sanger sequenced to confirm the fusion's presence in tumor tissue and absence from adjacent normal tissue.

## Western blotting

293T cells were selected as a surrogate model and grown in RPMI with 10% fetal bovine serum (FBS). Constructs (pcDNA and pcDNA-LPCAT1-TERT-Flag) were reverse-transfected using FuGene (Promega) following the manufacturer's protocol. 30,000 cells were seeded in a 6cm plate. The ratio of FuGene reagent to DNA was 3:1. Cells were harvested 48hrs later. Anti-Flag antibody (Sigma, mouse, 1:1000) was used to determine the expression level of transfected constructs, and anti-tubulin antibody (Sigma, T5026, mouse, 1:3000) was used as a control. Further details on fusion construct generation are included in the S1 Methods.

## Cell culture conditions and cell growth assay

The HEK293G were obtained from American Type Culture Collection (ATCC) (Manassas, VA). DMEM was used for the cultures under standard incubation conditions of 37C and 5% CO2 were used in all experiments. Purchased media was enriched with 10% fetal bovine serum. Transfection was performed as described above except that 5,000 cells were seeded per well on 96-well plates, and 5 replicates were made in each condition. On reverse transfection, medium with 10% FBS was used, and at 72 hours post-transfection, 1% of resazurin solution at 0.1µg/µl (Sigma-Aldrich) was added in each well. Plates were incubated at 37˚C for 4 hours and the fluorescent signal was measured at 560Ex/590Em wavelength, in a FL X800 Microplate Reader (BIO-TEK Instruments Inc).

## Cellular localization

Coverslips (25mm circle) were placed in 6 well plates and plated with HEK293 cells at a 100,000 cells per well. Cells were transfected with 2ug of DNA using X-tremeGENE™ HP (Roche cat. 06366546001). At 48 hours post transfection cells were rinsed with PBS and fixed with 4% formaldehyde for 15 minutes, rinsed again with PBS, and permeabilized for 10 minutes using 0.1% Triton X-100 in PBS. Cells were then blocked for 30 minutes in 1% BSA, 22.52mg glycine in PBST (0.1% Tween 20). Blocking solution was removed and cells were incubated overnight at 4˚C in rabbit Anti-Flag (Sigma-Aldrich, cat. F7425) at a 1:200 dilution in PBST with 1% BSA. Cells were rinsed with PBS and secondary antibody, goat anti-rabbit Alexa-488 (Invitrogen, cat. A11008), was added at 1:500 dilution in PBST with 1% BSA and incubated at room temperature in the dark for 1 hour. Coverslips were rinsed with PBS then mounted using ProLong™ Gold Antifade with DAPI and allowed to set overnight. Images were obtained using Zeiss LSM 800 confocal microscope and processed using ImageJ software.

## Results

### *LPCAT1-TERT* fusion transcripts occur in epithelioid trophoblastic tumors

Nine cases of GTD with sample qualities yielding sufficient DNA and RNA to enable testing were identified from internal and external treating centers. Samples were collected for three ETTs, four PSNs (one atypical) and two PSTTs (Table 1). ETT-2 originated at Maternity Hospital Kuwait. ETT-1 and PSTT-2 originated at Mayo Clinic while all other successfully profiled samples were acquired through Mayo Clinic's consultation practice. Available clinicopathologic data were limited due to the fact that the majority of cases were clinical consults. Four further ETTs from Johns Hopkins University School of Medicine produced DNA and RNA yields insufficient to proceed with analysis and were excluded from the study. Representative histologic images are provided in Fig 1.

RNA-Seq and fusion transcript detection were performed on all samples in Table 1 with the exception of the atypical PSN case which was not available (NA) at the time of testing. RNA-

**Table 1. Clinicopathologic details.**

| Case | Age at Diagnosis (years) | Diagnosis | Site | Specimen Received | Tumor Size (cm) | Additional Treatment and Follow-up |
|---|---|---|---|---|---|---|
| ETT-1 | 47 | ETT | Uterine corpus and cervix (primary) and liver (metastasis) | Hysterectomy, BSO (primary), liver resection (metastasis) | UNK (primary), 8 (metastasis) | Completed 3 cycles of chemotherapy; developed multiple metastases (liver, spleen, lungs, brain) 3 years later and died of disease |
| ETT-2 | 51 | ETT | Uterine fundus | Hysterectomy, BSO | 5 | UNK |
| ETT-3 | 46 | ETT | Endometrium | Endometrial biopsy | UNK | UNK |
| PSN-1 | 36 | PSN | Endometrium | Endometrial polypectomy and curettage | UNK | UNK |
| PSN-2 | 26 | PSN | Endometrium | Endometrial curettage | UNK | UNK |
| PSN-3 | 34 | PSN | Endometrium | Hysterectomy, BSO | UNK | UNK |
| APSN-1 | 43 | Atypical PSN | Uterus, NOS | Submucosal lesion excision | UNK | UNK |
| PSTT-1 | 41 | PSTT | Uterus, NOS | Hysterectomy, BSO | 3 | UNK |
| PSTT-2 | 31 | PSTT | Uterus, NOS, with adnexal soft tissue extension | Hysterectomy, BSO | 6 | Completed 5 cycles of chemotherapy; developed pulmonary metastases 2 years later and is undergoing immunotherapy |

ETT = epithelioid trophoblastic tumor; PSN = placental site nodule; PSTT = placental site trophoblastic tumor; UNK = unknown; BSO = bilateral salpingo-oophorectomy; NOS = not otherwise specified

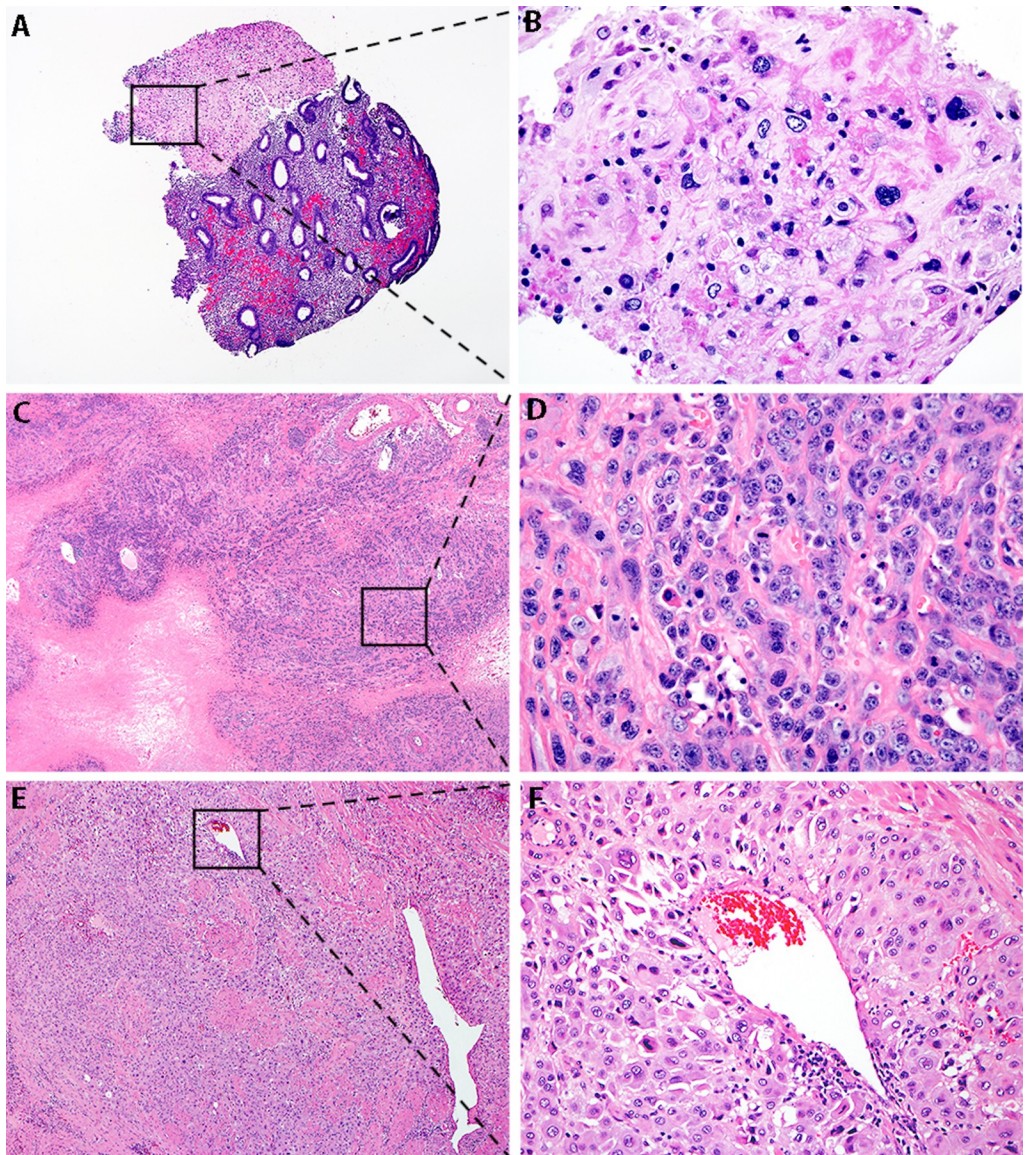

**Fig 1. Representative histologic images.** Magnification 40x (A, C, E) and 200x (B, D, F). (A, B) Placental site nodule. Well-circumscribed nodular lesion composed of chorionic-type intermediate trophoblast with abundant clear to eosinophilic cytoplasm and round nuclei, embedded in a hyalinized matrix. (C) Epithelioid trophoblastic tumor. Large expansile nests and nodules, separated by eosinophilic hyaline-like material composed of chorionic-type intermediate trophoblast. The trophoblast have a moderate amount of clear to eosinophilic cytoplasm and relatively uniform round nuclei with small nucleoli and multiple mitotic figures (D). (E, F) Placental site trophoblastic tumor. Infiltration of myometrium by sheets of implantation site-type intermediate trophoblast composed of large cells with abundant eosinophilic to amphophilic cytoplasm with pleomorphic nuclei, some of which are seen in association with the wall of a blood vessel.

Seq indicated the presence of *LPCAT1-TERT* fusion transcripts in two of three ETTs tested (Table 2). All other samples tested were negative for *TERT* fusions on the basis of RNA-Seq analysis. No control samples showed any evidence of *LPCAT1-TERT* fusion-supporting reads. *LPCAT1* and *TERT* are colinear transcripts separated by an approximately 165kb genomic region that contains *SLC6A3* and *CLPTM1L* on the reverse strand of chromosome 5. Both ETTs with putative *LPCAT1-TERT* fusions were predicted to produce multiple splice variants

**Table 2. Junction exon combinations, genomic coordinates, reading frame status and supporting read counts for *LPCAT1-TERT* fusions identified in two epithelioid trophoblastic tumors by RNA-Seq.**

| Case | Coordinates (hg19) | Exon Numbers (*LPCAT1-TERT*) | Reading Frame Preserved? | Total # Supporting Reads |
|------|-------------------|------------------------------|--------------------------|--------------------------|
| ETT-1 | Chr5:1501576–1282739 | Exon 2 –Exon 3 | No | 71 |
| | Chr5:1501576–1294781 | Exon 2 –Exon 2 | No | 69 |
| | Chr5:1494815–1282739 | Exon 3 –Exon 3 | Yes | 136 |
| | Chr5:1494815–1294781 | Exon 3 –Exon 2 | No | 380 |
| | Chr5:1494811–1282739 | Intron 3 –Exon 3 | No | 304 |
| ETT-2 | Chr5:1523825–1294781 | Exon 1 –Exon 2 | Yes | 303 |
| | Chr5:1523825–1282739 | Exon 1 –Exon 3 | No | 227 |

with one transcript in each case predicted to produce an intact protein product on the basis of a preserved reading frame (Figs 2 and 3).

## *LPCAT1-TERT* fusions are rarely reported and non-recurrent in other cancers

*LPCAT1-TERT* fusions were confirmed absent from all internal and public normal and tumor tissue databases profiled (see Methods). Literature review identified only three previous reports of *LPCAT1-TERT* fusions, each identified in distinct neoplasms affecting neurological, liver, and lung tissue respectively [27–30] (Table 3).

## Copy-number changes underlie *LPCAT1-TERT* fusions and *TERT* upregulation in ETT

Whole genome DNA copy number array analysis was performed for all samples to determine the possibility of a genomic deletion affecting the region between the fused exons due to *LPCAT1*'s genomic positioning upstream of *TERT*. ETT-1 demonstrated three to four copy number gain of chromosome 5 but also showed reduced probe intensities consistent with low-level loss corresponding to 3' *LPCAT1*, 5' *TERT*, and all intervening genes. These findings were classified as supportive of an approximately 200kb genomic deletion. ETT-2 also appeared to have a three to four copy number gain of chromosome 5 as well as reduced probe intensities supportive of a two-copy genomic deletion underlying the formation of the *LPCAT1-TERT* fusion. Collectively these results support genomic deletions underlying the formation of the *LPCAT1-- TERT* fusions and while the array-based copy number analysis lacks the resolution to determine precise genomic breakpoints, the formation of multiple splice forms of the fusion transcript and their joining at precise exon boundaries are indicative of intronic breakpoints. ETT-3 and all other samples were classified as negative for genomic deletions affecting *LPCAT1* or *TERT*, however ETT-3 was classified as demonstrating single copy number gain of chromosome 5 including *TERT* and *LPCAT1*. PSN-3 could not be assessed due to sample quality issues. ETT-1 and ETT-2 each demonstrated copy number gain of chromosomes 2, 3, 9 and 20, with loss of chromosome 11. ETT-3 showed gain of chromosomes 5, 9 and 20 in common with the other two ETTs. All non-ETT cases were categorized as normal in terms of gross copy number (S1–S8 Figs).

## *TERT* expression levels are elevated in ETT compared to other GTDs and normal tissue

*LPCAT1* and *TERT* expression levels for all GTD cases are provided in Table 4. *TERT* gene expression levels were universally elevated in ETT when compared to other GTDs and normal tissues. The mean normalized count was 7219.28 in ETT, 24.167 in PSN, 8.332 in PSTT and

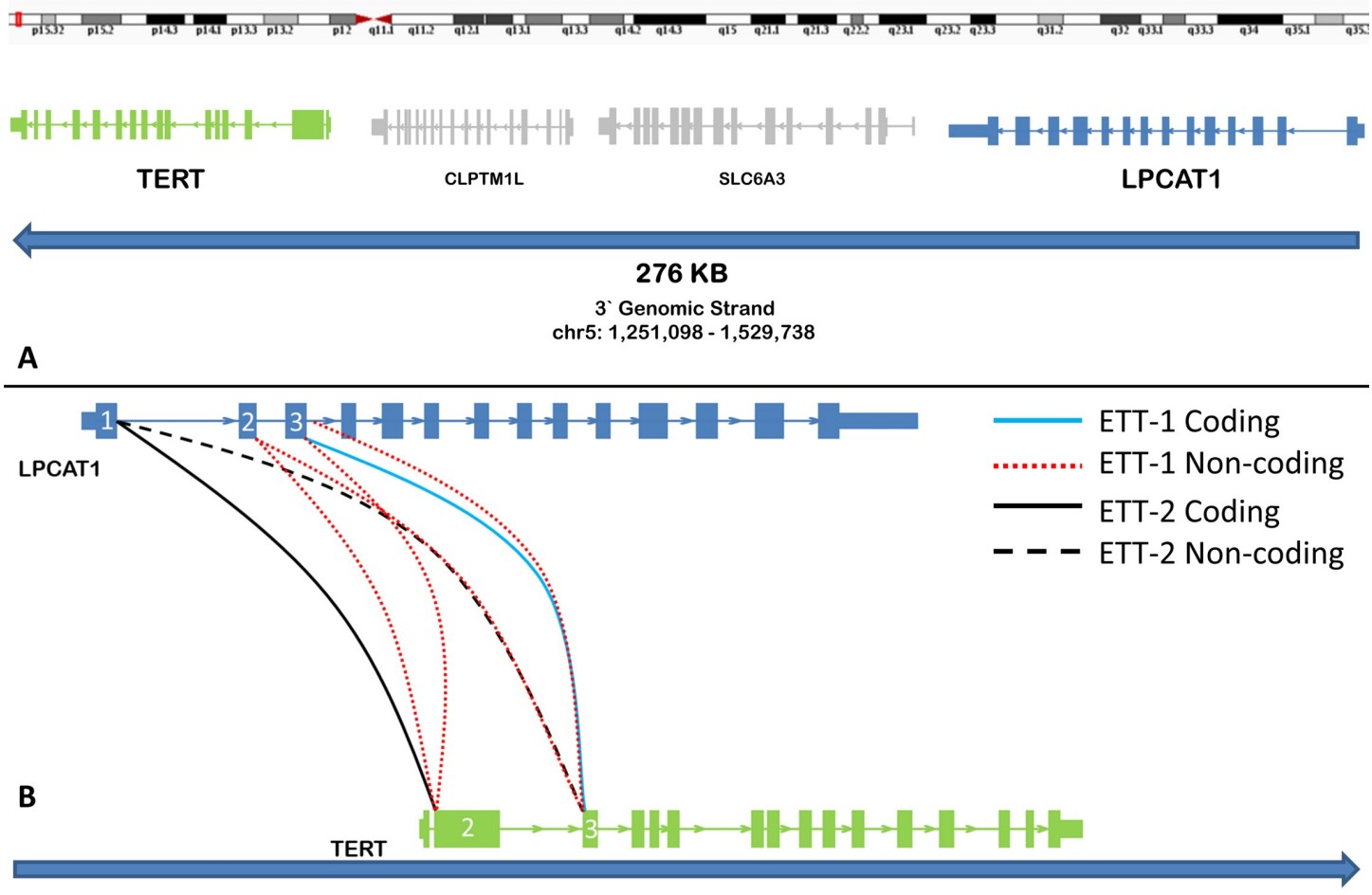

**Fig 2. *LPCAT1-TERT* fusion transcripts identified in two epithelioid trophoblastic tumors.** A) The genomic location of *LPCAT1* and *TERT* on chromosome 5 are displayed. Both genes are colinear and are transcribed from the reverse genomic strand. *LPCAT1* lies upstream of *TERT* with two intervening genes positioned between them. B) Alternative *LPCAT1-TERT* transcript isoforms identified in the two epithelioid trophoblastic tumor cases. Orientation has been flipped from (A) for readability. Solid lines denote exon combinations that retain reading frame and are likely to result in a translated protein product while dotted lines indicate abrogation of reading frame. Both ETT1 and ETT2 produce one transcript that is predicted to form a protein coding *LPCAT1-TERT* fusion transcript. Distinct exon combinations are observed between the two tumors.

17.83 in PSTT plus PSN. *LPCAT1* gene expression levels in our cohort appeared elevated in ETT-2 and ETT-3 but low in ETT-1 relative to the other GTDs. Previously reported *TERT* expression levels (S9 Fig) in normal uterine (n = 142) and ovarian tissue (n = 180) as recorded by the GTEx initiative [26] were negligible (median 0.0 TPM). *TERT* promoter analysis was conducted for all samples to determine if alternative, established mechanisms of *TERT* activation might be present in any other case of GTD. All samples tested negative for known activating or novel *TERT* promoter mutations.

### *LPCAT1-TERT* fusion is an early event in metastatic ETT-1

The *LPCAT1-TERT* fusion event initially detected in case ETT-1 was identified in a liver metastasis that was resected three years after initial diagnosis and treatment of ETT (Table 1). Following detection of the fusion in the metastatic tumor, the primary tumor was tested for the presence of the in-frame fusion transcript. RT-qPCR and Sanger sequencing verified that the

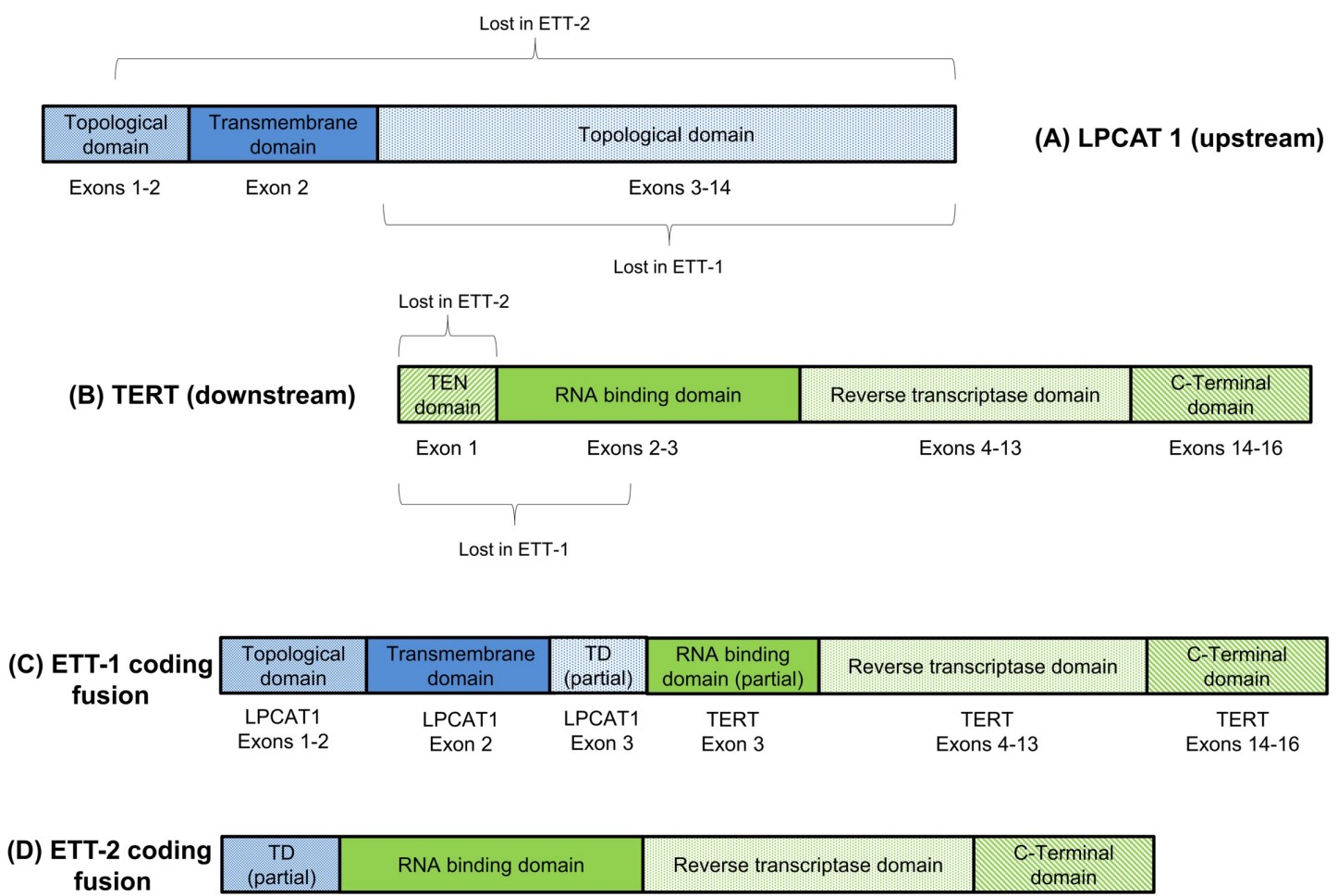

**Fig 3. Predicted preserved protein domains for the in-frame LPCAT1-TERT fusions in ETT-1 and ETT-2.** (A) Native LPCAT1 (upstream of TERT), and (B) TERT (downstream of LPCAT1) domains are illustrated in upper image, with the regions predicted to be lost in the fused products indicated by labeled brackets. Putative fusion proteins (C, D) are shown in the lower image. In each fusion, TERT lacks components (TEN domain and partial/whole RNA binding domain) believed critical for normal telomerase function. The LPCAT1 topological domain (TD) retained in ETT-1 and partially retained in ETT-2 is cytoplasmic in nature while the transmembrane domain retained in ETT-1 only is helical. A fragment of the larger (lumenal) LPCAT1 topological domain is retained in ETT-1 and lost in ETT-2. Regions are not drawn to scale.

primary tumor sample carried an *LPCAT1-TERT* fusion identical to that originally identified in the metastatic tissue, while adjacent normal tissue showed no evidence of the fusion (Fig 4).

**Table 3. Previously reported LPCAT1-TERT fusions and presence in current cohort.**

| Reference | Fusion observed | Observed in current cohort? | Protein coding or non-coding | Tissue |
|---|---|---|---|---|
| [27] Manuscript body | LPCAT1 exon 11 upstream of TERT exon 2 | No | Coding | Meningioma |
| [28] S5 Table | LPCAT1 exon 1 upstream of TERT exon 2 | Yes (ETT-2) | Coding | Lung adenocarcinoma |
| [29] Conference abstract only | Unspecified LPCAT1-TERT fusion | Unknown | Coding | Hepatocellular carcinoma |
| [30] Database compiled from multiple sources. | LPCAT exon 1 upstream of TERT exon 3 | Yes (ETT-2) | Non-coding | Lung adenocarcinoma |

**Table 4. *LPCAT* and *TERT* expression levels (DESeq2 normalized counts) for all GTD cases.**

| Case | *LPCAT* Expression | *TERT* Expression |
|------|------|------|
| ETT1 | 6284.8 | 4523.2 |
| ETT2 | 16478 | 16648 |
| ETT3 | 15874 | 486.64 |
| PSN1 | 2410.7 | 4.5032 |
| PSN2 | 2173.8 | 62.397 |
| PSN3 | 3465.5 | 5.5998 |
| APSN1 | NA | NA |
| PSTT1 | 2307.6 | 16.661 |
| PSTT2 | 5697.2 | 0 |

## LPCAT1-TERT fusion protein positively modulates cell growth and localizes predominantly in the nucleus

To define the functionality of this in-frame fusion, 293T cells were selected as a surrogate cell-line and transfected with pcDNA control vector or pcDNA-LPCAT1-TERT-Flag, and cell viability was determined after 72 hours by the fluorimetric indicator dye resazurin. The pcDNA-LPCAT1-TERT-Flag cells showed a significant increase in viability compared to the pcDNA control measured by the metabolic capacity (Fig 5A). Expression of LPCAT1-TERT fusion protein was confirmed by Western Blot using 293T cells transfected with pcDNA control vector or pcDNA-LPCAT1-TERT-Flag and anti-flag antibody or anti-tubulin as control (Fig 5B). As expected, TERT and LPCAT1 were mostly localized in the nucleus and cytosol, respectfully. TERT-LPCAT1 fusion localization was mostly observed in nucleus with some localizing in the cytosol (S10 Fig).

## Discussion

We have described the first instance of reoccurring genomic events in ETT and demonstrated positive regulatory effects on cell growth in a surrogate cell-line. The discovery of an *LPCAT1--TERT* fusion transcript in ETT marks the first report of protein-coding *LPCAT1-TERT* fusion recurrence in any tumor type, with only four prior published reports of similar fusions occurring sporadically in other neoplasms [27–30]. Fusion events involving alternative gene partners fused with either *LPCAT1* or *TERT* are also rare, evidenced by the presence of only three *LPCAT1* and twenty-four *TERT* fusions in a public database of samples from The Cancer Genome Atlas [25], further underscoring the novelty of this discovery in a rarely observed neoplasm.

Exons 1–3 of *LPCAT1* encode two topological domains (a full cytoplasmic domain and a partial lumenal domain) and one helical transmembrane domain. It is uncertain whether these bestow unique function on the fusion, beyond the increased transcriptional activity of the downstream *TERT* exons under promoter control of the ubiquitously expressed *LPCAT1*. However, the observance of some cytosolic localization of the fusion protein product may be the result of the retained cytoplasmic topological domain. The TERT protein normally possesses 4 domains believed critical for telomerase function, comprising the Ten domain, RNA-binding domain, reverse-transcriptase domain, and C-terminal domain [31]. The Ten domain is encoded by exon 1 while exons 2 and 3 encode the RNA binding domain. Thus, the protein-coding fusions detected in both ETT-1 and ETT-2 will exhibit ablation of the Ten domain, while ETT-1 will also partially lack the RNA-binding domain, indicating that the growth modulating functions likely occur independently of telomerase function. Ideally future studies could be expanded to include trophoblast-specific cell lines, in order to further clarify the

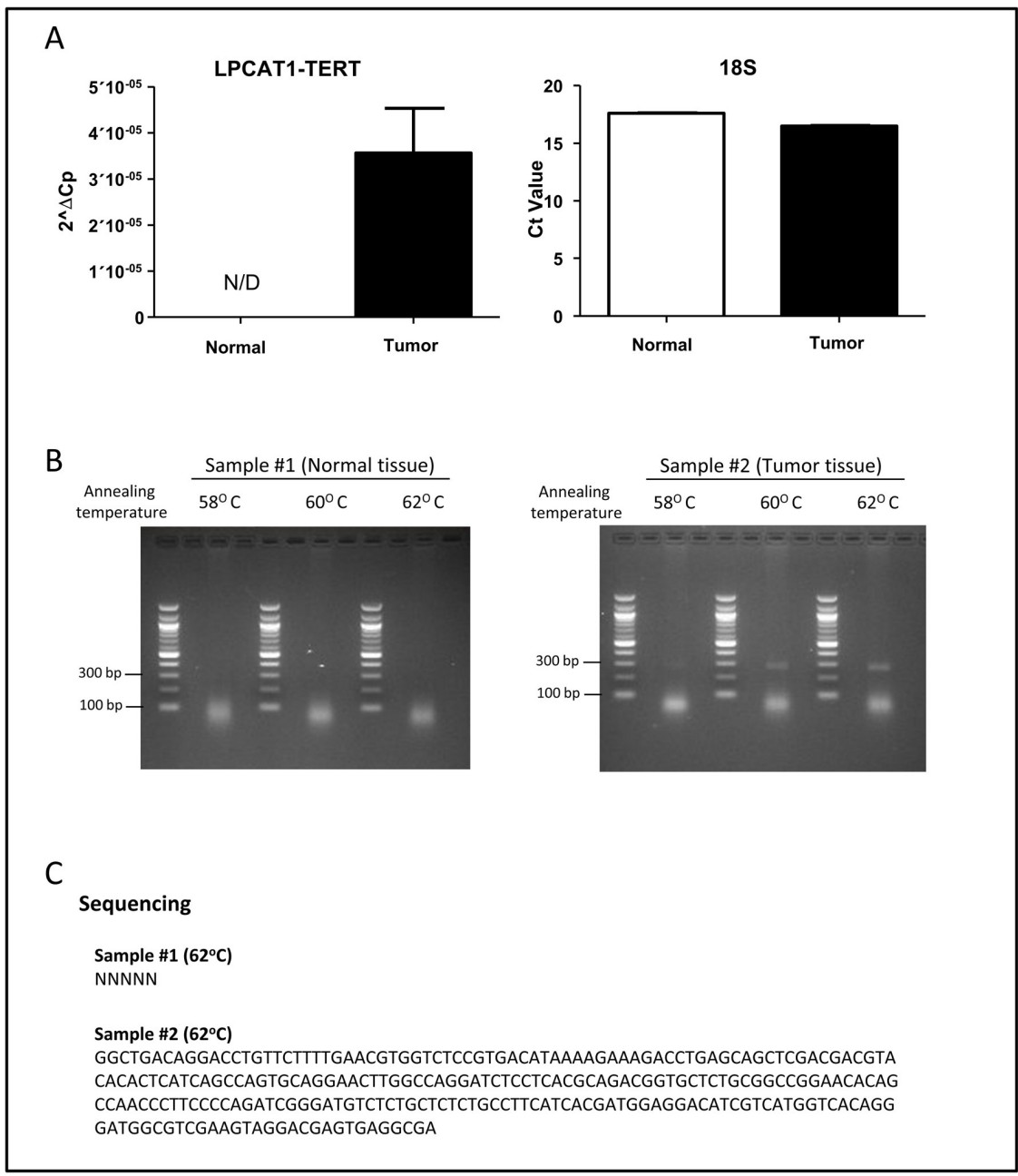

**Fig 4. Confirmed expression of *LCPAT1-TERT* fusion transcript in ETT-1 primary tumor and absence from adjacent normal tissue.** A) Real-time PCR quantification of the fusion in normal and tumor tissues in adjacent normal and primary tumor tissue utilizing an 18S RNA control. B) Gel electrophoresis of the PCR product from the tumor tissue. C) Sanger sequencing result for the PCR product produces a chimeric *LPCAT1-TERT* transcript. The originally tested sample for case ETT-1 was a liver metastasis occurring at relapse 3 years post-surgery and treatment. Confirmation of the fusion transcript in the primary tumor indicates that *LPCAT1-TERT* formation was an early event in the disease pathogenesis.

precise biological role of *LPCAT1-TERT* fusions in trophoblastic tissue. For instance, a TRAP assay might be utilized in the presence and absence of *LPCAT1-TERT* to experimentally validate the absence of telomerase activity. Whether predisposing genomic factors underlie the genomic deletions that create *LPCAT1-TERT* fusions remains unknown and represents another avenue of future research.

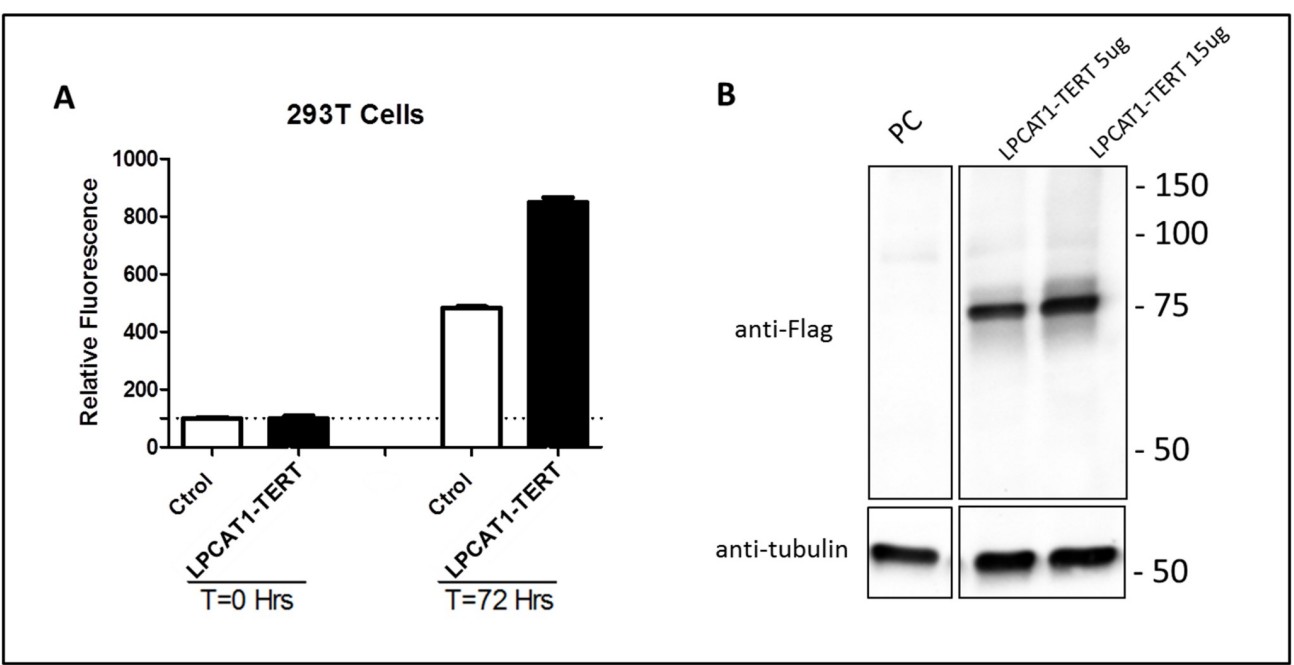

**Fig 5. *LPCAT1-TERT* fusions positively regulate cell growth.** A) Cell viability measured at 0 and 72 hours utilizing fluorimetric indicator dye resazurin. Increased metabolic capacity of pcDNA-LPCAT1-TERT-Flag cells demonstrate increased viability (sample size = 5). B) Western Blot confirms expression of LPCAT1-TERT fusion protein in 293T cells using 293T cells transfected with pcDNA control vector or pcDNA-LPCAT1-TERT-Flag and anti-flag antibody or anti-tubulin as control (PC = pcDNA control vector).

Non-telomerase-based oncogenic functions of TERT are widely supported. While its onco-genicity is most commonly attributed to cellular immortalization through telomerase-driven telomere extension, multiple alternative mechanisms of action have been established [32, 33]. Perhaps most remarkably, one previous study demonstrated that alternatively spliced *TERT* transcripts affected by a ten-exon deletion retained the ability to stimulate cell proliferation through activation of Wnt signaling [34]. Irrespective of precise function, the fusing of *LPCAT1* and *TERT* is expected to bring the fused *TERT* exons under the transcriptional control of the more ubiquitously utilized *LPCAT1* promoter and is considered *TERT* activating. It is possible that the *LPCAT1* overexpression in ETT-2 and ETT-3 might play a distinct role in pathogenesis, while wild-type *TERT* expression likely also plays a part since copy number gain and increased transcription were observed in all ETTs regardless of the fusion's presence. However, the precise interplay of factors is difficult to unravel without further study.

*TERT* fusions have previously been suggested to represent a rare mechanism of *TERT* activation in cancer [18]. However, the most frequently reported means of *TERT* upregulation is promoter mutation in the form of activating single nucleotide variants [35]. No evidence of *TERT* promoter mutations was identified in any of the GTDs tested in our study. This fact, together with the elevated expression of *TERT* in ETT compared to a near-complete absence of expression in the other GTDs, suggests both that *TERT* activation may be unique to ETT versus the other tested forms of GTD, and that gene fusion may represent a common means of activation. Telomerase activity has previously been reported in choriocarcinoma and in hyda-tidiform moles where it has been associated with persistent or metastatic disease [36–40], but to our knowledge, no study has assessed its activity in ETT, PSN or PSTT, nor demonstrated underlying genomic alterations. It is possible that *TERT* upregulation might be an early event in ETT and that *LPCAT1-TERT* fusion might represent a later event capable of driving disease

aggressiveness as has been observed for alternative *TERT* promoting events in tumor types
[41] where an *LPCAT1-TERT* fusion has been reported [27]. Based on current evidence and
limited clinical information, however, it is not possible to make specific conclusions about the
sequential timing of *LPCAT1-TERT* fusions or their exact role in disease presentation or pro-
gression. Only partial information regarding disease presentation and prior pregnancy status
were available. While case ETT-1 was known to have given normal birth to an only child 18
years prior to primary tumor diagnosis and had no clinically recorded history of molar preg-
nancy, clinical information available for ETT-2 made no mention of normal or molar preg-
nancies, and no prior information regarding ETT-3 was available. Furthermore, while case
ETT-1 was known to suffer recurrence of disease, nothing is known of the other two cases as
both were clinical consult cases which were unfortunately lost to clinical follow-up.

The chance recurrence of protein coding *LPCAT1-TERT* fusions in two of three tested
ETTs is improbable—an assertion supported by the existence of no more than a single prior
report of a similar fusion within approximately ten thousand Cancer Genome Atlas samples
[30] and only three previous reports throughout the published literature [27–29]. Beyond
recurrence, the early emergence of the fusion in the tested primary tumor and its demon-
strated growth promoting effects support a potentially important role in the oncogenic pro-
cesses of ETT. With further research, we believe it is likely that *LPCAT1-TERT* fusions will be
identified in further ETTs. If this is indeed proven, then the likelihood of clinical utility will
increase.

The discovery of copy number-driven *TERT* upregulation and *LPCAT1-TERT* fusions in
ETT indicates potential diagnostic or prognostic relevance for a disease lacking unambiguous
markers. While morphologic examination or immunohistochemical profiling often allow
pathologists to accurately diagnose gestational trophoblastic neoplasms, screening of GTD
samples for *TERT* expression or the presence of the *LPCAT1-TERT* fusion*s* (either RNA or
protein) may offer greater diagnostic certainty in differentiating ETT from histologically simi-
lar lesions, especially when immunohistochemical markers overlap. In this study, aneuploidy
was a characteristic uniquely observed in ETT, with several shared whole chromosomal gains
and losses amongst the three cases. This in itself may offer diagnostic or prognostic utility,
however prior studies have been lacking and contradictory [42–44]. The *LPCAT1-TERT* fusion
might provide prognostic indications if demonstrated to be restricted to aggressive or recur-
rent forms of the disease. Our findings in ETT also raise the prospect of novel treatment
modalities. Increasingly TERT is being targeted by traditional and nascent antineoplastic
approaches [45], and some or all of these might represent therapeutic possibilities, since the
*LPCAT1-TERT* fusion has been shown to be growth promoting. While the unique nature of
the fusion might render it untargetable with existing treatments, its protein coding nature
raises the possibility of susceptibility to novel therapeutic agents. Novel neo-antigens produced
by aberrant proteins in cancerous cells are increasingly being targeted by modern immuno-
therapy-based initiatives [46] and could offer an area of future research in fusion-expressing
ETTs. However, the precise nature and extent of the effects of targeting the fusion in ETT cells
remains to be explored.

While this manuscript was being finalized, a separate study applying RNA-Seq to ten cases
of GTD was published [47]. While this new study reported a pathogenic *PIK3CA* mutation in
one case of ETT, and inferred activation of the *PIK3CA* pathway based on transcriptome-wide
differential expression analysis, no recurrent genomic events were reported in four ETTs or six
other cases of GTD, despite fusion transcript analysis being performed. What the findings of
that study indicate in unison with our own remains uncertain. No *PIK3CA* mutations were
identified by inspection of raw RNA data for the three ETT cases included in our study, and
transcriptome-wide differential expression analysis was not performed due to the difficulty in

producing reliable global expression comparisons between heavily degraded samples with variable processing times and conditions. Fusion transcript detection is often hindered by fusion calling algorithms' default filter settings as we have previously described [48]. Indeed, the *LPCAT1-TERT* fusion described in this study was initially removed by TopHat Fusion's default filtering settings and was only discovered by manual analysis of filtered results, necessitating the custom approach described in our Methods. Unfortunately, the data of Cho *et al.* are not publicly available to enable our reanalysis.

Both sample availability and quality pose significant obstacles to research of GTDs and in particular ETTs. This study increases the published number of ETTs by two, since case ETT-2 was previously published as a case report [49]. The total number of ETTs described in the literature likely now approaches only 140 cases since their first description in 1998, and the number of samples in our study matches or exceeds that of previous molecular studies involving GTD [42–44, 47], indicative of their rarity. In total, we sourced seven cases of ETT, but unfortunately four were degraded to the extent that neither sequencing nor copy number analysis were possible. While testing was possible for the remaining three cases, samples were exhausted by the molecular profiling described, meaning that further molecular characterization of these samples is not possible, despite further study being of potential value. While disease rarity cannot be overcome, the issues of sample quality and quantity can theoretically be addressed. Procurement of fresh-frozen rather than FFPE samples, for example, offers a means of circumventing such issues. However, it is logistically difficult due to established clinical practices.

In conclusion, we have identified *LPCAT1-TERT* fusion transcripts and copy number-driven *TERT* upregulation as characteristics of ETTs that appear to be absent from PSNs and PSTTs. These findings have potential diagnostic, prognostic and therapeutic relevance, the extent of which we hope will be elucidated through further studies. Sample quality and rarity pose challenges to expanded testing, however awareness of the findings described here should enable rapid and targeted testing of new ETT candidates and add valuable knowledge to our clinical and biological characterization of ETTs, and GTD in general.

## Supporting information

**S1 Fig. Chromosomal microarray performed on DNA extracted from FFPE tissue using the OncoScan CNV Plus assay for ETT-1.** ETT-1 demonstrated gain of chromosome 5 but also showed reduced probe intensities consistent with low-level loss corresponding to 3'*LPCAT1*, 5'*TERT*, and all intervening genes.
(PPTX)

**S2 Fig. Chromosomal microarray performed on DNA extracted from FFPE tissue using the OncoScan CNV Plus assay for ETT-2.** ETT-2 appeared to have gain of chromosome 5 as well as reduced probe intensities supportive of a two-copy genomic deletion underlying the *LPCAT1-TERT* fusion.
(PPTX)

**S3 Fig. Chromosomal microarray performed on DNA extracted from FFPE tissue using the OncoScan CNV Plus assay for ETT-3.** ETT-3 was classified as negative for genomic deletions affecting *LPCAT1* or *TERT*, but demonstrated copy number gain of chromosome 5.
(PPTX)

**S4 Fig. Chromosomal microarray performed on DNA extracted from FFPE tissue using the OncoScan CNV Plus assay for PSN-1.** PSN-1 was classified as negative for copy number

alterations and genomic deletions affecting *LPCAT1* or *TERT*.
(PPTX)

**S5 Fig. Chromosomal microarray performed on DNA extracted from FFPE tissue using the OncoScan CNV Plus assay for PSN-2.** PSN-2 was classified as negative for copy number alterations and genomic deletions affecting *LPCAT1* or *TERT*.
(PPTX)

**S6 Fig. Chromosomal microarray performed on DNA extracted from FFPE tissue using the OncoScan CNV Plus assay for PSN-3.** PSN-3 could not be assessed due to sample quality issues.
(PPTX)

**S7 Fig. Chromosomal microarray performed on DNA extracted from FFPE tissue using the OncoScan CNV Plus assay for PSTT-1.** PSTT-1 was classified as negative for copy number alterations and genomic deletions affecting *LPCAT1* or *TERT*.
(PPTX)

**S8 Fig. Chromosomal microarray performed on DNA extracted from FFPE tissue using the OncoScan CNV Plus assay for PSTT-2.** PSTT-2 was classified as negative for copy number alterations and genomic deletions affecting *LPCAT1* or *TERT*.
(PPTX)

**S9 Fig. Expression of *TERT* in normal tissues in GTEx Release 8.** Ovarian and Uterine tissues are marked with a red asterisk (Median TPM of 0.0).
(PPTX)

**S10 Fig. Representative localization Immunohistochemistry images of HEK293 cells expressing empty vector, *TERT*, *LPCAT1*, *TERT-LPCAT1-fusion* with a C-term 3xFlag. Anti-Flag (green), DAPI (blue).** TERT-LPCAT1 fusion is observed to localize mainly in the nucleus with some cytosolic localization (sample size = 2).
(PPTX)

**S1 Raw images.**
(PDF)

**S1 Methods.**
(DOCX)

## Author Contributions

**Conceptualization:** Gavin R. Oliver, Jonathan Quist, Eric W. Klee.

**Data curation:** Gavin R. Oliver, Sofia Marcano-Bonilla.

**Formal analysis:** Gavin R. Oliver, Jonathan Quist, Naresh Prodduturi, Numrah Fadra, Michael Zimmerman, Jan B. Egan.

**Funding acquisition:** Eric W. Klee.

**Investigation:** Gavin R. Oliver, Sofia Marcano-Bonilla, Amy A. Swanson, Raul Urrutia, Ema Veras, Rema'a Al-Safi, Matthew Block, Sarah Kerr, Martin E. Fernandez-Zapico, John K. Schoolmeester.

**Methodology:** Gavin R. Oliver, Eric W. Klee.

**Software:** Gavin R. Oliver.

**Supervision:** Gavin R. Oliver, Amy A. Swanson, John K. Schoolmeester, Eric W. Klee.

**Validation:** Ezequiel J. Tolosa, Eriko Iguchi, Nicole L. Hoppman, Tanya Schwab, Ashley Siga-foos, Jesse S. Voss, Shannon M. Knight, Jin Zhang, Raul Urrutia, Anthony G. Bilyeu, Jin Jen, Martin E. Fernandez-Zapico.

**Visualization:** Gavin R. Oliver.

**Writing – original draft:** Gavin R. Oliver.

**Writing – review & editing:** Gavin R. Oliver, Amy A. Swanson, Matthew Block, Sarah Kerr, Martin E. Fernandez-Zapico, John K. Schoolmeester, Eric W. Klee.

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
