## [Decision Letter · Decision Letter 0]

10 Feb 2021

PONE-D-20-36897

LPCAT1-TERT Fusions Are Uniquely Recurrent in Epithelioid Trophoblastic Tumors and Positively Regulate Cell Growth

PLOS ONE

Dear Dr. Oliver,

Thank you for submitting your manuscript to PLOS ONE. I am very sorry for the delay in reaching a decision on your manuscript entitled “LPCAT1-TERT Fusions Are Uniquely Recurrent in Epithelioid Trophoblastic Tumors and Positively Regulate Cell Growth" due in part to difficulty in securing reviewers with appropriate expertise to comment upon all aspects of your work in a reasonable time. We have now received reports from 3 reviewers and, after careful consideration, we have decided to invite a minor revision of the manuscript.

After careful consideration, we feel that it has merit but does not fully meet PLOS ONE’s publication criteria as it currently stands. Therefore, we invite you to submit a revised version of the manuscript that addresses the points raised during the review process.

We look forward to receiving your revised manuscript.

Kind regards,

Bruno Bernardes de Jesus

Academic Editor

PLOS ONE

Journal Requirements:

2.Please provide additional details regarding participant consent. In the ethics statement in the Methods and online submission information, please ensure that you have specified (1) whether consent was informed and (2) what type you obtained (for instance, written or verbal, and if verbal, how it was documented and witnessed). If your study included minors, state whether you obtained consent from parents or guardians. If the need for consent was waived by the ethics committee, please include this information.

3.Thank you for including the following ethics statement on the submission details page:

'The study was reviewed and approved by the Mayo Clinic Institutional Review Board.'

Please also include this information in the ethics statement in the Methods section of your manuscript.

4.PLOS ONE now requires that authors provide the original uncropped and unadjusted images underlying all blot or gel results reported in a submission’s figures or Supporting Information files. This policy and the journal’s other requirements for blot/gel reporting and figure preparation are described in detail at https://journals.plos.org/plosone/s/figures#loc-blot-and-gel-reporting-requirements and https://journals.plos.org/plosone/s/figures#loc-preparing-figures-from-image-files. When you submit your revised manuscript, please ensure that your figures adhere fully to these guidelines and provide the original underlying images for all blot or gel data reported in your submission. See the following link for instructions on providing the original image data: https://journals.plos.org/plosone/s/figures#loc-original-images-for-blots-and-gels.

5.Thank you for stating the following in the Acknowledgments Section of your manuscript:

"The Genotype-Tissue Expression (GTEx) Project was supported by the Common Fund of the Office of the Director of the National Institutes of Health, and by NCI, NHGRI, NHLBI, NIDA, NIMH, and NINDS."

Reviewers' comments:

Reviewer's Responses to Questions

**Comments to the Author**

1. Is the manuscript technically sound, and do the data support the conclusions?

Reviewer #1: Partly

Reviewer #2: Yes

Reviewer #3: Yes

2. Has the statistical analysis been performed appropriately and rigorously? 

Reviewer #1: Yes

Reviewer #2: N/A

Reviewer #3: Yes

3. Have the authors made all data underlying the findings in their manuscript fully available?

Reviewer #1: No

Reviewer #2: Yes

Reviewer #3: Yes

4. Is the manuscript presented in an intelligible fashion and written in standard English?

Reviewer #1: Yes

Reviewer #2: Yes

Reviewer #3: Yes

5. Review Comments to the Author

Reviewer #1: Thank you for an interesting manuscript. I have a few major and minor points that I believe need to be addressed prior to publication. I would firstly kindly request the authors to publish all sequencing and microarray data on a public site such as Sequence Read Archive or European Nucleotide Archive with data release following successful publication of the manuscript. Please include the SRA/ENA identifier in the manuscript in a section dedicated to data availability.

My other point is around the "so what" of the article. The authors report two cases of LPCAT1-TERT fusions in a rare cancer type. While it is certainly interesting that a recurring alteration can be identified, the authors admit that no other group, even in the same tumour type, has identified this fusion. Therefore what would line of sight for, say, clinical development look like? Is there a population or are these two cases one-off? Please discuss in the article.

If one were to design compounds against the fusion protein, looking at Cansar https://cansarblack.icr.ac.uk/target/Q8NF37/synopsis/ligandability there doesn't seem to be any evidence for ligandability for LPCAT1. At the same time, the Cancer Dependency Map illustrates that LPCAT1 isn't essential in almost any cell line (https://dmc.depmap.org/portal/gene/LPCAT1?tab=overview) so as a target it might have few toxicity issues in normal tissues. Would the authors suggest targeting TERT instead? I didn't fully get it from the article but have you shown that knocking down or out the fusion protein is lethal for the cells, such that this is not just a passenger event?

Bullet point list of other points

- Regarding presence of this fusion in other data sets, see https://fusionhub.persistent.co.in/out/global/Individual/LPCAT1--TERT.html indicating one in a TCGA LUAD sample (out-of-frame)

- Are these fusions intronic at the DNA level? Whole genome or targeted DNA sequencing of the two genes including intronic tiling should reveal real breakpoints far better than RNA-seq and if this fusion indeed results from a deletion (or, say, tandem duplication).

- Table 4, while the effect size difference is quite considerable, RPKM mustn't be used to compare across samples. Use DESeq2 or similar on counts to normalise data prior to comparisons instead or discuss why DESeq2 cannot be used in this case.

- Figure 2 What does the fusion look like for domains that remain from both partners? Is LPCAT1 replacing some kind of a regulating mechanism of TERT? A genome browser such as https://lifescience.opensource.epam.com/ngb/index.html visualises fusion junctions and which domains remain. Please include the domains and discussion about their significance.

- "Unfortunately, the data of Cho et al. are not publicly available to enable our reanalysis, and attempts to contact the authors have been unsuccessful." Could you please remove the passive aggressive tone; please merely state that the data is not available for reanalysis.

- Using the unfiltered bioinformatics approach risks false positives. Did you apply the same unfiltered approach to unrelated samples to see a base level of false positives to rule out chance finding?

I hope you find these comments useful for your work.

Reviewer #2: Epithelioid trophoblastic tumor (ETT), a rare form of gestational trophoblastic disease (GTD), shows high risk of metastasis at the time of diagnosis and corresponding mortality. Through the profiling of nine cases of GTD comprising ETT, PSTT and PSN, Lysophosphatidylcholine acyltransferase (LPCAT1), LPCAT1-TERT fusion transcripts were identified that appear to uniquely reoccur in ETT and are caused by genomic deletions.

LPCAT1-TERT fusion expression was validated by RT-qPCR.

DNA mutation analysis of TERT promoter region was conducted for all samples.

Copy number gains of chromosome 5 were shown to accompany TERT upregulation in ETT, even in the absence of LPCAT1-TERT fusion.

1. As indicated by the authors, LPCAT1-TERT fusion and TERT promoter mutations have been previously found in rare instances in meningiomas by Juratli et al. (Oncotarget 8, 109228-109237, 2017), hepatocellular carcinoma by Haines et al. (Cancer Research abstract A33, 2016), and lung adenocarcinomas (CHIMERSEQ).

However, this is the first report on occurrence in ETT. Further interest is provided by its early occurrence in primary tumor and persistence in metastatic tissue.

2. LPCAT1-TERT fusion proteins were shown to promote cell growth, but the mechanism was indicated most likely unrelated to telomere extension. However, wild-type TERT expression was suggested to play a part since copy number gain and increased transcription were observed in all ETTs regardless of the fusion’s presence. TERTp mutations are among the most common recurrent alterations in human cancer. However, no evidence of TERT promoter mutations was identified in any of the GTDs tested.

Hence, further study is needed to solve this issue. However, this article provides relevant guidance.

Minor issue

This reviewer fully understands the ‘defensive nature’ of some of the comments by the authors in the Discussion, e.g. the lack of reply by colleagues working in the field. However, this is unnecessary in a final text. The limitations of the study (sample number, rarity of the disease, limited understanding of its patho-physiology) should be summarized as such in one paragraph.

Reviewer #3: The paper is very interesting, and the work quite well done. There are only a few minor aspects to address to facilitate readers comprehension of the text and one experiment proposed that would help understand the role of the LPCAT1-TERT construct.

---

## [Author Response · Author response to Decision Letter 0]

26 Mar 2021

Dear Editor and Reviewers,

We would like to express our gratitude to all involved for their thoughtful and diligent review of our manuscript entitled “LPCAT1-TERT Fusions Are Uniquely Recurrent in Epithelioid Trophoblastic Tumors and Positively Regulate Cell Growth”. We are grateful for your decision to accept the manuscript for publication in PLOS ONE pending minor revisions and clarifications. Below we address each of the several points raised specifically by the reviewers and editor. Note that provided line numbers refer to the version of the revised manuscript with tracked changes.

Editor’s comments:

We have reviewed the style requirements and have made changes as appropriate i.e.

- Corrected capitalization in manuscript title and section headers

- Corrected figure headers i.e. Fig 1 vs Figure 1

- Removed underscores from figure file names

- Removed funding information from Acknowledgements 

- Corrected format of author affiliations

- Corrected format of corresponding author listing

2. Please provide additional details regarding participant consent. 

Additional details have been provided within an ethics statement in the Methods section, and in the online submission.

3. Thank you for including the following ethics statement on the submission details page:

'The study was reviewed and approved by the Mayo Clinic Institutional Review Board.' Please also include this information in the ethics statement in the Methods section of your manuscript.

This has been completed, as explained above. 

4. PLOS ONE now requires that authors provide the original uncropped and unadjusted images underlying all blot or gel results reported in a submission’s figures or Supporting Information files. 

The original, uncropped and unadjusted images have been provided in file S1_raw_images.pdf and this has been detailed in the cover letter.

5. Please remove any funding-related text from the manuscript and let us know how you would like to update your Funding Statement. Currently, your Funding Statement reads as follows: "The author(s) received no specific funding for this work." Please include your amended statements within your cover letter; we will change the online submission form on your behalf.

We have provided a new funding statement for publication and have removed any reference to funding from the manuscript. “We wish to acknowledge Mayo Clinic Center for Individualized Medicine and the Department of Laboratory Medicine and Pathology for supporting this study. GTEx data used for the analyses described in this manuscript were obtained from dbGaP accession number phs000424.v7.p2. The Genotype-Tissue Expression (GTEx) Project was supported by the Common Fund of the Office of the Director of the National Institutes of Health, and by NCI, NHGRI, NHLBI, NIDA, NIMH, and NINDS.”

Reviewer 1:

1. I would firstly kindly request the authors to publish all sequencing and microarray data on a public site such as Sequence Read Archive or European Nucleotide Archive with data release following successful publication of the manuscript. Please include the SRA/ENA identifier in the manuscript in a section dedicated to data availability.

Copy number array data have been deposited in ArrayExpress with accession E-MTAB-10303 while sequencing data have been deposited under accession E-MTAB-10321. Data will be published upon successful publication of our manuscript. We have detailed this in the Additional Information section of the submission form, as required by PLOS One.

2. The authors report two cases of LPCAT1-TERT fusions in a rare cancer type. While it is certainly interesting that a recurring alteration can be identified, the authors admit that no other group, even in the same tumour type, has identified this fusion. Therefore what would line of sight for, say, clinical development look like? Is there a population or are these two cases one-off? Please discuss in the article.

We thank the reviewer for raising this very relevant issue. While the findings of this study alone are insufficient to conclude certain clinical benefits, the discovery of LPCAT1-TERT fusions in two out of three tested ETT samples, and its absence from other GTDs and other tumors in general is unlikely to occur by chance. If we consider that only one LPCAT1-TERT fusion is known to have been identified through multiple analyses of approximately 10,000 TCGA samples, as compared to 2 of 3 of our tested ETTs, the observed enrichment corresponds to an odds ratio of 5407 and a Fisher’s p-value of 5.992 x10-7. Given further samples and available testable material, we believe it is likely that similar fusions could be identified in further ETTs. It is also possible that retesting of the scarce, published ETT RNA-Seq could yield positive findings since fusion detection is highly sensitive to differing filtering strategies. If further LPCAT1-TERT fusions are indeed identified ETTs, the opportunity to further explore diagnostic or therapeutic options grows. Unfortunately, we are currently unable to prove these theories conclusively, but we hope our findings will empower future researchers to pursue them successfully. We have included new text to this effect in the Discussion at lines 377 onwards.

3. If one were to design compounds against the fusion protein, looking at Cansar there doesn't seem to be any evidence for ligandability for LPCAT1. At the same time, the Cancer Dependency Map illustrates that LPCAT1 isn't essential in almost any cell line so as a target it might have few toxicity issues in normal tissues. Would the authors suggest targeting TERT instead? I didn't fully get it from the article but have you shown that knocking down or out the fusion protein is lethal for the cells, such that this is not just a passenger event?

Thanks to the reviewer for raising well-considered points. Further research is required to determine if a knockdown of the fusion would negatively impact cell proliferation or be lethal to ETT cells, however multiple factors (recurrence, emergence in primary tumor and demonstrated growth-promoting effects) provide strong indication that the fusion is more than a passenger event (we have added language to this effect at line 377 onward) and targeting of the fusion from therapeutic standpoint would represent a viable research avenue in the event of further confirmation of the fusion’s recurrence and functional relevance. We raise the prospect of treatment options, largely pertaining to targeting of TERT or the fusion itself at line 393 onwards but further work will be required to explore the possibility of such approaches and we have added clarifying language to this effect at line 397 onward.

4. Regarding presence of this fusion in other data sets, see https://fusionhub.persistent.co.in/out/global/Individual/LPCAT1--TERT.html indicating one in a TCGA LUAD sample (out-of-frame)

We thank the reviewer for providing this example. We have confirmed the identification of a non-coding LPCAT1-TERT fusion in TCGA data and have included a suitable reference in Table 3 and made minor edits to the main text where required. We have furthermore updated Table 3 to include the tissue of origin and coding status of previously identified LPCAT1-TERT fusions. We hope that these edits address the reviewer’s comment adequately. 

5. Are these fusions intronic at the DNA level? Whole genome or targeted DNA sequencing of the two genes including intronic tiling should reveal real breakpoints far better than RNA-seq and if this fusion indeed results from a deletion (or, say, tandem duplication).

We thank the reviewer for this question and commentary. The Oncoscan array-based copy number analysis included in the manuscript indicates that the cause of the fusion is indeed genomic deletion. Unfortunately, the array probe density is not sufficient to identify precise breakpoints however the fact that we see multiple splice variants of the RNA fusions is strongly indicative of intronic breakpoints and subsequent splicing. We have added language to the Results section of the manuscript (line 259 onward) to highlight this. While an alternative genomic methodology could offer the ability to identify breakpoints with higher resolution, this is currently not possible in the context of this study due to the exhaustion of genetic material.

6. Table 4, while the effect size difference is quite considerable, RPKM mustn't be used to compare across samples. Use DESeq2 or similar on counts to normalise data prior to comparisons instead or discuss why DESeq2 cannot be used in this case.

We thank the reviewer for their insight. We have subsequently removed RPKM-based expression from the manuscript and have recalculated expression levels for LPCAT1 and TERT utilizing DESeq2 and its median of ratios approach. We have added language in the Methods to describe this and have updated Table 4 and the Results section (line 272 onward) as well as adding an appropriate reference. We hope that these amendments address the reviewer’s concerns.

7. Figure 2 What does the fusion look like for domains that remain from both partners? Is LPCAT1 replacing some kind of a regulating mechanism of TERT? Please include the domains and discussion about their significance.

We thank the reviewer for highlighting an important aspect of the paper. The domains and promoter effects are included in the Discussion section (line 331 onward), and we have added to this as well as creating a new figure and legend specifically dedicated to the predicted domains involved in the fusion proteins (Fig 3). We opted for a custom-drawn figure to ensure we optimally represent each fusion based on what information is available in databases like as well as the published literature. We hope that these collectively address the reviewer’s concerns. 

8. "Unfortunately, the data of Cho et al. are not publicly available to enable our reanalysis, and attempts to contact the authors have been unsuccessful." Could you please remove the passive aggressive tone; please merely state that the data is not available for reanalysis.

We have edited the text as suggested by the reviewer to ensure it is not interpreted as passive-aggressive in tone. 

9. Using the unfiltered bioinformatics approach risks false positives. Did you apply the same unfiltered approach to unrelated samples to see a base level of false positives to rule out chance finding?

We thank the reviewer for outlining a very valid consideration. We routinely check against control data generated from tens of thousands of samples as outlined in the Methods. All control databases were negative for samples/reads supporting LPCAT1-TERT fusions. We have added text making this clear at line 209 of the Results.

Reviewer 2:

1. This reviewer fully understands the ‘defensive nature’ of some of the comments by the authors in the Discussion, e.g. the lack of reply by colleagues working in the field. However, this is unnecessary in a final text. The limitations of the study (sample number, rarity of the disease, limited understanding of its patho-physiology) should be summarized as such in one paragraph.

We thank the reviewer for their insight and comments. We have subsequently amended the statement regarding the data of Cho et al to state “Unfortunately, the data of Cho et al. are not publicly available to enable our reanalysis”, which we hope fully addresses the related comments and suggestions provided by Reviewer 1 and Reviewer 2. The difficulties and limitations of the study are summarized in a single paragraph and we have edited the language in order to remove a defensive tone (line 416 onward).

Reviewer 3:

1. The fusion LPCAT1-TERT is a key part of the work. These genes and their main functions should be clarified in introduction. In fact, the paragraph written in discussion (lines 270 to line 275) could perfectly fit in the “Introduction section”.

We agree fully with this statement and have moved the described paragraph to the Introduction of the manuscript at line 72 onward. Further discussion of precise domains has also been included in the Discussion at lines 331 onward.

2. Figure 1 would be much easier to understand if authors would use color arrows (or other symbols) to indicate the factors described in the figure legend 1 (i.e. nodules, expansile nests, etc.) Also, include a bar with the magnification of the image.

Thanks to the reviewer for this suggestion. Figure 1 has now been updated to outline the magnified regions of the image, which are described in the figure legend. Magnification levels have been added to the figure legend.

3. Legend of Figure 4. write what “n” indicates.

We have edited the figure legend (Fig 4 is now Fig 5) and replaced “n” with “sample size”.

4. Line 284. “Furthermore, while assessment of telomere length in the presence or absence of LPCAT1-TERT has the potential to experimentally validate the absence of telomerase activity…”. In many tumors, telomerase is activated but tumor cells have short telomeres, which are just minimally maintained so cells can continue to divide. The best evidence for telomerase activity is TRAP assay, which could be developed in a cell line which does not have telomerase activity, after LPCAT1-TERT transfection.

We than the reviewer for their insight. We have edited the corresponding part of the Discussion at lines 334 onward to include mention of the TRAP assay and we hope that our edits will adequately address this comment. 

5. Since authors have the LPCAT-TERT-Flag construct, it would be interesting to know the localization cellular localization of the fusion protein, even if it were done in the 293T cells. 

We thank the reviewer for their suggestion. We have subsequently conducted new experimentation to show nuclear localization of the LPCAT1-TERT fusion protein. We have included a new supplementary figure (S10 Fig) with legends and corresponding language in the Methods (line 166), Results (line 305) sections and Discussion (line 334). 

6. Authors find fusions between exons 1-3 of LPCAT1 and exons 1-2 of TERT gene. Is there any reason why these genomic regions are more prone to fusions than other regions?

We are unaware of any factor predisposing these locations to breakage or joining. This is perhaps not surprising, since while the events are recurrent, they are nonetheless extremely rare and this means they are unlikely to correspond to e.g. known fragile regions or topologically associated domains for example. We have included language to this effect in the Discussion at line 345 onward and hope that we have satisfactorily addressed the reviewer’s question.

7. Figure 2. The dotted line for ETT1 non-coding fusion is not easily distinguishable from the solid line. Perhaps authors could use another color to make it more visible.

Thanks to the reviewer for this observation. We have changed the colors of the solid and dotted lines for ETT-1 in Figure 2 to aid with differentiation. We hope that this addresses the reviewer’s concerns.

8. Please, write “NA stands for not available” underneath Table 3. Alternatively, make it clear in the text (line 172): “not available (NA)”. 

Thanks to the reviewer for highlighting this issue. As suggested, we have added “not available (NA)” to the main text to aid with clarity. 

9. Just for clarity, in Figure 4 authors could write in the legend that Cp stands for “Crossing point” or whatever it may mean.

Thanks to the reviewer for identifying this oversight. We have included needed clarity here by adding text to the figure legend (Fig 4 is now Fig 5) to indicate that PC indicates the pcDNA control vector. 

Thanks once more to all who have dedicated their time and attention to the improvement of our manuscript. We hope that we have addressed these points to the satisfaction of the reviewers and reviewing editor. If further information or clarification is required, we will be happy to promptly respond.

Best Regards,

Gavin Oliver

---

## [Decision Letter · Decision Letter 1]

8 Apr 2021

LPCAT1-TERT Fusions Are Uniquely Recurrent in Epithelioid Trophoblastic Tumors and Positively Regulate Cell Growth

PONE-D-20-36897R1

Dear Dr. Oliver,

We’re pleased to inform you that your manuscript has been judged scientifically suitable for publication and will be formally accepted for publication once it meets all outstanding technical requirements.

Kind regards,

Bruno Bernardes de Jesus

Academic Editor

PLOS ONE

Additional Editor Comments (optional):

Reviewers' comments:

Reviewer's Responses to Questions

**Comments to the Author**

1. If the authors have adequately addressed your comments raised in a previous round of review and you feel that this manuscript is now acceptable for publication, you may indicate that here to bypass the “Comments to the Author” section, enter your conflict of interest statement in the “Confidential to Editor” section, and submit your "Accept" recommendation.

Reviewer #1: All comments have been addressed

Reviewer #2: All comments have been addressed

Reviewer #3: All comments have been addressed

2. Is the manuscript technically sound, and do the data support the conclusions?

Reviewer #1: Yes

Reviewer #2: Yes

Reviewer #3: Yes

3. Has the statistical analysis been performed appropriately and rigorously? 

Reviewer #1: Yes

Reviewer #2: Yes

Reviewer #3: Yes

4. Have the authors made all data underlying the findings in their manuscript fully available?

Reviewer #1: Yes

Reviewer #2: Yes

Reviewer #3: Yes

5. Is the manuscript presented in an intelligible fashion and written in standard English?

Reviewer #1: Yes

Reviewer #2: Yes

Reviewer #3: Yes

6. Review Comments to the Author

Reviewer #1: (No Response)

Reviewer #2: I am pleased to see that all requests have been satisfactorily addressed. The best of luck for a high impact of your article on readers.

Reviewer #3: The authors have addressed all the issues raised by this reviewer. This reviewer is fully satisfied.

Just check line 306. There are two words repeated “in the in the nucleus”

7. PLOS authors have the option to publish the peer review history of their article (what does this mean?). If published, this will include your full peer review and any attached files.

Reviewer #1: **Yes: **Miika Ahdesmaki

Reviewer #2: **Yes: **Saverio Alberti

Reviewer #3: **Yes: **María Elisa Varela Sanz (E. Varela in publications)

---

## [Editor Report · Acceptance letter]

14 May 2021

PONE-D-20-36897R1 

*LPCAT1-TERT* fusions are uniquely recurrent in epithelioid trophoblastic tumors and positively regulate cell growth 

Dear Dr. Oliver:

I'm pleased to inform you that your manuscript has been deemed suitable for publication in PLOS ONE. Congratulations! Your manuscript is now with our production department. 

Kind regards, 

on behalf of

Dr. Bruno Bernardes de Jesus 

Academic Editor

PLOS ONE